# Offices after the COVID-19 Pandemic and Changes in Perception of Flexible Office Space

**Matus Barath** [1,*] and **Dusana Alshatti Schmidt** [2]

1   Department of International Management, Faculty of Management, Comenius University, 820 05 Bratislava, Slovakia
2   Faculty of Management, Comenius University, 820 05 Bratislava, Slovakia
*   Correspondence: matus.barath@fm.uniba.sk

**Abstract:** The pandemic is fast-moving, accelerating rapid changes that lead to new challenges and impacting organizations. A big mark has been left on the workplaces—places where we do business—because the ongoing change to remote work challenges the role of the office. It is highly possible that as the change progresses, the workplace will not only change its design but also the way in which work will be planned, organized, performed and controlled. However, as the restrictions ease up, questions arise: What is the potential of office sustainability? How has the perception of flexible office space changed due to the COVID-19 pandemic? This paper used an online survey as a quantitative research method. In this paper, we looked at the employer's vision of the office. We investigated employers' perspectives of where and in what settings the work will be performed in the post-pandemic time. Specifically, we discussed the changes employers will apply in terms of the work environment and office layout. The findings suggest that an increasing mobile workforce and expansion of the new work style will not mean an office exodus but will certainly have an impact on office utilization.

**Keywords:** work environment; employers; office space; remote work; COVID-19

## 1. Introduction

The office environment has always been seen as a main driver of organizational strategy, culture and performance and has been an integral part of a control function [1]. Over the past decades, the rising demand for flexibility has transformed traditional office environment toward flexible alternatives. Flexible office space (open plan, activity-based, co-working spaces, satellite office, home office and other remote locations) can be characterized as a dynamic work environment intended to be used in combination with flexible working arrangements. Hence, flexible office spaces provide employees with a range of ways and places to work [2] and are often associated with sustainable development goals [3]. Such work settings allow internal and external collaboration [4], knowledge sharing [5] and support innovation [6].

The development of the COVID-19 pandemic in 2019 led most organizations worldwide to implement remote work (work from home, home office) policies as a response to the government guidelines and regulations that were intended to maintain the health and safety of employees [7]. This induced a massive shift of employees from the office space to the home environment [8], either full-time or as a blend of home and in-office work (hybrid work model). As a result, the office space merged with the personal space. However, it was an absolute necessity and the only way to keep some departments in operation at all [9]. Whereas the physical interaction was unavailable, the digital transformation allowed employees to collaborate in a virtual workspace despite located in distinct place [10,11].

Under the unusual circumstances, the remote work experience has had both positive and negative impacts on employees. Many employees have enjoyed and benefited from

an enforced work-from-home opportunity and have expressed a strong desire to continue working from home as a standard way of work in the future [12]. However, conducting work within the home environment, inadequate workspace, childcare and social isolation represent stressors that have impacted employees' overall performance [13,14]. Adequacy of workspace at home, along with the home office setup (technology–accountability, expenditures and maintenance) and compliance with health and safety standards have stimulated discussions of whether to support employees working from home in the medium to long term [15]. Many employers have already funded employees' home office setups. Hence, employers may believe that working from home will become more common after the pandemic [16]. However, not all employers can allow employees to work from home due to the nature of their job. In the USA, only 37% of jobs [17] can be performed from home. This presents a challenge to the physical layout of the office. In this case, organization with flexible office spaces (open plan, co-working or activity-based) must invest in redesign of offices to minimize the future virus transmission risk and resistance of employees returning to office.

Accordingly, the main objective of this research was to investigate the potential sustainability of offices and identify the willingness of managers to change or adapt to the demands in the post-pandemic era. The aim was to analyze the impact of the pandemic on the office space, assess the attitude of managers toward the changes associated with the flexible office space after the pandemic and propose solutions for more effective work environments in the post-pandemic time. This paper assesses the latter in the regions of Slovakia and Kuwait, countries with different socioeconomic development, yet bilateral relations.

The study aims to answer the following two primary research questions:

*RQ1: How will the work environment change in the near future?*

*RQ2: How will employers approach the implementation of changes in their offices?*

The study contributes to existing knowledge on flexible office spaces. The findings contribute to literature by identifying managers' attitudes and changes in the perception of office space before, during and after the COVID-19 pandemic from regions with lack of previous research on this topic. The study can benefit organizations as it provides effective solutions to office design considering diverse the post-pandemic needs of employees, which can lead to their increased satisfaction, motivation and productivity.

This paper is divided into five sections. Section 1 is the introduction. Section 2 presents literature review. In Section 3, research methods are described. In Section 4, the interpretation of the results, findings and discussion are explained. Section 5 is the conclusion, with research findings with limitations and future research directions.

## 2. Literature Review

### 2.1. Flexible Office Space and the Effects of the Pandemic

The flexible office space, also known as flexi-space, includes the fundamental features of the traditional office, such as desks, chairs, phones and computers. However, it provides dynamic environment that meets the needs of both employees and employers.

The open office is an office configuration with minimum or no interior boundaries, such as walls or partitions between employees. Its characteristic features are openness and flexibility [18,19]. The idea is to facilitate communication and idea flow in organizations [20]. Open offices were very popular before the pandemic, mainly for managers. Conversely, employees have attempted to adjust to such office design. During the pandemic, such office design represented high infection risk due to the crowded environment and minimum distance between desks. Therefore, employers must pay attention to personal space and apply strict health and safety measures [21].

The co-working office is where a group of different employees, usually employed by different employers or self-employed, use a common, shared workplace. Such spaces provide the opportunity for remote workers to work in a more stimulating environment rather than home [22]. However, as a result of the pandemic, co-working spaces must adapt their operations and rethink the office layout to ensure that governmental restrictions and

preventive measures can be more effectively met. Building virtual co-working and digital co-working mechanisms that keep the community together can help employees feel more secure and minimize the adverse impact of the home office [23].

The activity-based office allows employees to work on various activities, whether alone or in collaboration with others, while changing multiple types of flexible work settings during the day with no fixed desk in the workplace [24]. Haapakangas et al. [25], in their study, indicated that an active use of activity-based offices is associated with productivity and well-being at work due to satisfaction with privacy, communication and the physical environment in general. However, the work time spent on searching for workspace was detrimental to both outcomes. Therefore, when striving for the improved productivity and well-being of employees, employers should prioritize privacy, communication and smooth workspace transition. To date, there has been no research related to the effects of the COVID-19 pandemic on the activity-based office.

The satellite office is a type of remote work environment where an employee works in a center established by the employer outside the employer's premises, e.g., at the client's place in their location or region. The implementation of such offices helps organizations sustain employees' health and safety during the unprecedented situation, such as the COVID-19 pandemic, by minimizing the size of the office, hence minimizing the probability of infection and office closure. Assigning employees to an office nearby can save their daily commuting time, thus protecting them from the virus. Furthermore, it decreases the company's carbon footprint. It will also enable the company to recognize social responsibilities for the neighborhood and community to contain the spread of the virus during the pandemic [26].

The home office became an important part of the pandemic. Close to 40% of those currently working in the EU began to work from home full-time [27]. Work from home gives employees the flexibility to arrange their work time and environment. Whereas work-life balance and satisfaction with this form of work increased during the pandemic [28], overall environmental discomfort resulting from home office distractions, inadequate tools, the presence of family members or stress and anxiety caused by social isolation led to inability to concentrate [29–31]. Consequently, productivity decreased. Yang et al. [28] highlighted the importance of home-based work environments and organizational supports for work-from-home arrangements in the post-pandemic era.

Other remote locations (offices) are alternative workspaces or locations that are open to the public, such as cafés, libraries, vacation homes, streets, parks, car parks or railway stations. This form of office is associated only with certain jobs and is usually tied to a good quality Internet connection [32].

*2.2. The Studied Countries*

Slovakia, as one of the 27 European Union countries with a population of over 5 million people, is considered a high-income advanced economy. The leading service sector with the highest contribution to GDP employs majority of the active population. Trade, real estate, tourism and banking industry dominate the sector. The industry sector, with high-value added industries (petrochemical, electronics, engineering, automotive and manufacturing), is the secondary contributor to GDP and employs more than one-third of the workforce. The agricultural sector is the least developed [33].

Relevant research on the topic of office layout was conducted before COVID-19. Barath [34] dedicated his dissertation research to the flexible office space in Slovakia. The results showed that the most popular space among the interviewed employers was the open concept office. Traditional and closed offices were mostly used in industry and production. Open offices were dominant mainly in services related to finance, accounting, banking, insurance and consulting. Shared offices or a combination of offices were most widespread in services related to IT, consulting and telecommunications.

According to Knapková [35], open offices in Slovakia have been on the rise. Hence, increasing demand for health and safety in the workplace during the pandemic increased

costs for employers, as office re-design was necessary. Furthermore, despite legal regulation of certain flexible work arrangements (part-time, telecommuting, homework and job sharing), their use is limited. This has led to amendments in the labor law at of beginning of the pandemic.

Kuwait is one of the Gulf Cooperation Council (GCC) countries with a high-income oil-based economy and population of over 4 million people. The oil industry, followed by the service sector (mainly financial services government, transportation and other private services), lead the economy and employ majority of employees. The non-oil industry (manufacturing and construction) and agricultural sectors are narrowed [36]. The country's economic growth is limited due to the demographic structure and dependence on the foreign labor force. Job security and well-paid employment benefit Kuwaiti nationals in the public (government) sector, while rigid labor market regulations and the lack of required skills impact the private sector [37], resulting in an overstaffed public sector and high government expenditures. Policies such as "Kuwaitization," intended to minimize (low-skill and highly skilled) expatriates in the private sector and instead attract Kuwaiti nationals, have not been successful [38]. As stated by Al-Mutairi et al. [39], one of the reasons are long working hours. It is noteworthy that despite the gradual flexibility of the labor markets in the GCC countries, the Kuwaiti Labor Code does not legislate flexible work arrangements except fixed-term contracts.

The COVID-19 pandemic has required many governments to adjust labor regulations worldwide. In Kuwait, work in all ministries and the public sector was suspended [40]. The Government of Kuwait released a document, namely the "Remote Work Guideline for the Public Sector" [41]. A few weeks later, when the complete lockdown was applied, the work was also performed from home in the private sector where applicable [42]. Even with the five-phase plan of transition to normal life [43] and vaccination initiatives, employers have been slow to rebound due to the limited number of employees allowed in the workplace, hence utilizing hybrid working practices [44]. Such practice requires adjustments not only to working schedule but also to the office design.

Since scarce research exists on the implication of the pandemic measures on organizations in Slovakia and Kuwait, we decided to address this gap and analyze the office design was before the pandemic, how it has changed during the pandemic and how it will look like in the post-pandemic time in these two distinct regions.

### 3. Research Methodology

*Research Methods, Data Collection and Data Analysis*

This study is based on a quantitative approach. Reasoning methods, namely logical induction and deduction, were used to analyze the data and draw conclusion in way of the theoretical contribution for researchers, with practical implications in terms of the new flexible office space model and managerial implication for practical use. Comparison methods were properly developed in the literature review, which helped to complexly draw conclusions between the selected countries and connect them with other authors' contributions relevant to the COVID-19 pandemic.

A structured self-designed online questionnaire survey was used as an instrument to collect information and provide a clearer picture of the COVID-19 pandemic changes to the role of offices. The questionnaire survey was developed based on the studied literature and was divided into four sections (demographics, office layout before the pandemic, office layout during the pandemic and office layout preferences after the pandemic). For the relevance of the research and complete anonymity of the respondents [45], the questionnaire survey was administered through Google Forms. In order to eliminate common method biases, the questionnaire avoided leading questions that encouraged respondents to choose a particular answer. Moreover, the questionnaire included a variety of questions (open-ended, close-ended and scales) and minimized overly technical terminology. Finally, the questionnaire was prepared in English, Slovak and Arabic and distributed through different channels (email and LinkedIn messages) [45].

The selection of appropriate sampling methods requires a thorough evaluation of possibilities and advantages and disadvantages of individual methods, which have been described by the authors [27,46] in the studied literature. In many cases, authors have relied on a sample size or statistical methods [47], which, due to their low relevance, were not used in making recommendations and conclusions. The selection of a sample plays a crucial role in the quality of the research; hence, the sample was drawn from 50 private and public organizations in the dominant sectors in each country, Slovakia and Kuwait. Due to the size of the sample, which does not represent the whole population evenly, the non-random sampling method (also known as non-probabilistic sampling method) was used. The sample was too broad and provided an overview of solutions in various sectors, but based on these results, a general output for the given sector can be assumed, such as public administration and administration, or organizations focused on open offices.

The targeted respondents were managers representing the employers (organizations) who were addressed through email or through professional networks. Further, in the paper, respondents are defined as managerial representatives who represented the sample and answered the online questionnaire survey. Managers, as change agents, symbolize organizational changes and are the closest to the implementation of new processes [48].

The final sample size is summarized in Table 1. Four-point Likert scale questions were recorded and converted to numerical. We used the two-way Chi-square test in Table 2, when the first variable was answer to the question and the second was the country (Slovakia, Kuwait). Based on Cramér's V and Pearson's correlation coefficients, we found a medium dependence between the variables. Other potential dependency in all selected tables were rejected (when: $\chi^2 > \chi^2_\alpha$ (significant level $\alpha = 0.05$) and $p$-value $< \alpha$) for small or trivial dependency. Using MS Excel, the critical value $\chi^2_\alpha$ was calculated using the function CHISQ.INV. A $p$-value was calculated using the function CHITEST.

**Table 1.** Breakdown of gender and job title.

|  |  | Slovakia (n = 38) | | Kuwait (n = 43) | |
|---|---|---|---|---|---|
|  |  | **n** | **%** | **n** | **%** |
| Gender | Female | 21 | 55.3 | 9 | 20.9 |
|  | Male | 17 | 44.7 | 34 | 79.1 |
| Job Title | Manager | 14 | 36.8 | 21 | 48.9 |
|  | Senior Manager | 7 | 18.4 | 9 | 20.9 |
|  | Director | 16 | 42.1 | 7 | 16.3 |
|  | Owner | 0 | 0 | 2 | 4.6 |
|  | Others | 1 | 2.7 | 4 | 9.3 |

**Table 2.** Preferences of flexible office spaces in the post-pandemic time.

|  | Slovakia (n = 38) | | Kuwait (n = 43) | |  |  |
|---|---|---|---|---|---|---|
|  | **n** | **%** | **n** | **%** |  |  |
| Flexible office space—open office—cubicle with high partitions | 12 | 31.6 | 9 | 20.9 | Statistic $\chi^2$ | 6.48 |
| Flexible office space—open office—cubicle with low partitions (6–10 employees) | 3 | 7.9 | 7 | 16.3 | Critical value $\chi^2_\alpha$ | 9.49 |
| Flexible office space—open office—limited partitions (11+ employees) | 2 | 5.3 | 8 | 18.6 | $p$-value | 0.262 |
| Flexible office space—coworking office | 7 | 18.4 | 9 | 20.9 | Cramer's V | - |
| Flexible office space—satellite center | 4 | 10.5 | 2 | 4.7 | Pearson's C | - |
| Flexible office space—activity-based office | 10 | 26.3 | 8 | 18.6 |  |  |

Based on gender, the sample from Slovakia was relatively balanced compared to Kuwait. In Kuwait, there were only two women among the higher positions (higher than the manager), while in Slovakia and at the base of the sample, women predominated in higher positions. From the point of view of the size of the organization where the respondent worked, medium and large organizations were the majority represented in Slovakia (86.8%). In Kuwait, large organizations (37.2%) had the largest representation, and medium and small organizations were in approximately the same representation, followed by micro-organizations (27.9%). As for the sectors, in Slovakia, government and public services were largely represented. In the private sector, the information technology and services sector was highly represented. In Kuwait, oil and gas, healthcare, and financial services and insurance were largely represented. The impact of cultural differences could cause differences in performance across countries, but this factor was not the main subject of the research.

## 4. Results

### 4.1. Work Environment and Office Space before the COVID-19 Pandemic

Managers were asked if the employees had the flexibility to work from home or other remote locations before the pandemic. Descriptive data analysis was completed using rankings and percentages for scale as in Figure 1a.

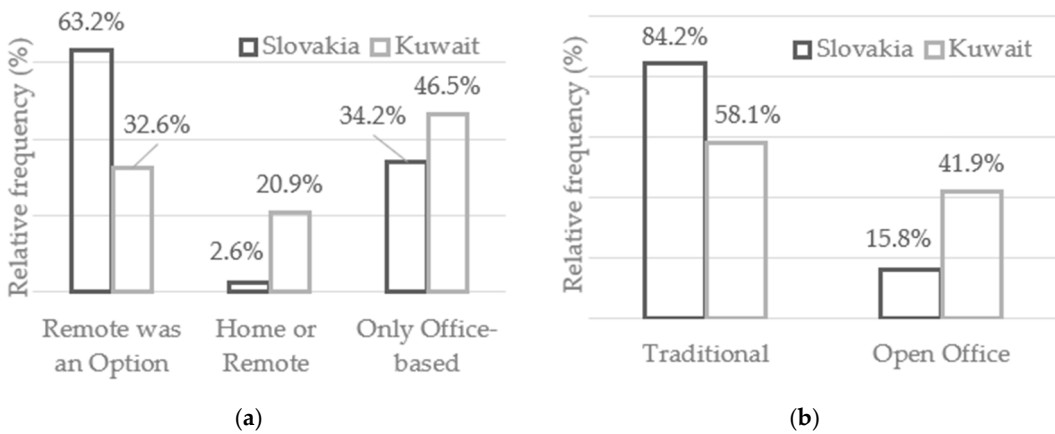

**Figure 1.** (**a**) Pre-pandemic: Work option, (**b**) pre-pandemic: office layout.

The work option was primarily office-based but working from home or another remote location was an option (63.2%) in Slovakia. The frequency was set at once a week or 5 times a month as a benefit. In comparison, 32.6% of employees in Kuwait had the opportunity to work from home or another remote location as an option. In total, 20.9% of employees in Kuwait worked from home or another remote location. The highest number of employees (46.5%) had only office-based work. Open offices were more adaptable and open for remote work. Based on the results from employers, the traditional arrangement of offices was more preferred in Slovakia. This phenomenon was obvious due to the nature of the work of individual departments. The results showed more balanced responses in the case of Kuwait, where the traditional arrangement of offices was preferred, but only in 58% of employers' responses. There was a benefit from a presumption in excess flexible time employment as teleworking open-share.

How important was the following for the purpose of the physical office before the pandemic?

(1) Building community and corporate culture
(2) Increasing productivity of employees
(3) Collaboration and socialization of employees
(4) Providing space for meetings with clients
(5) Providing learning and career development opportunities

(6)    Providing access to equipment and documents
(7)    Onboarding new hires
(8)    Innovating products or services
(9)    Attracting, retaining and nurturing talents

This question examined differences in perception of the physical office in Slovakia and Kuwait. On a scale from not important to very important, employers in Slovakia in Figure 2a reported higher importance differences in building community and corporate culture, enabling collaboration and socialization of employees compared to Kuwait in Figure 2b. Employers in the category of attracting, retaining and nurturing talents in Kuwait reported lower or negative importance.

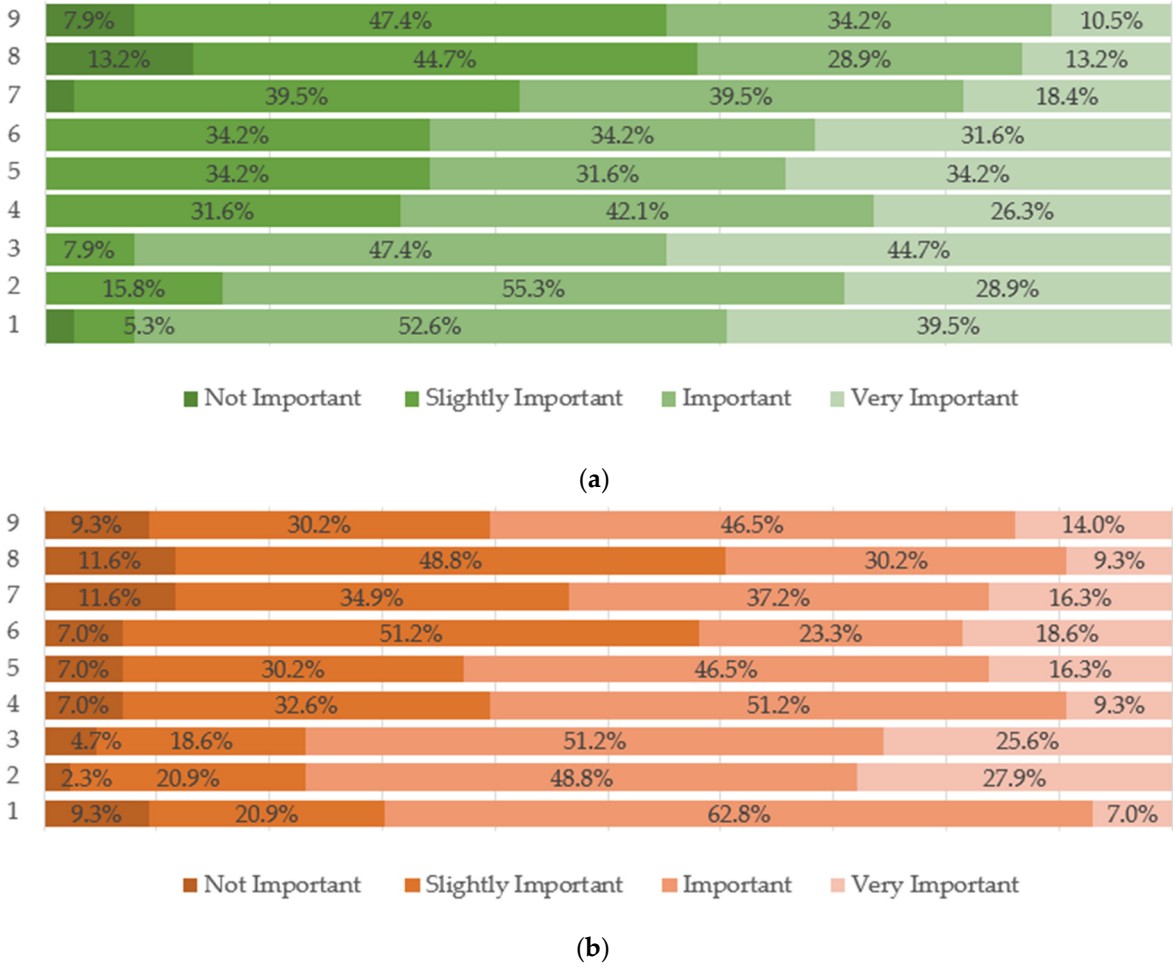

**Figure 2.** (**a**) Importance in selected categories (Slovakia), (**b**) importance in selected categories (Kuwait).

## 4.2. Work Environment and Office Space during the COVID-19 Pandemic

Managers were asked about changes caused by the pandemic. In Slovakia, 82% of employees worked remotely during the pandemic full or almost full-time as Figure 3 showed. In this case, the office layout played no role during the pandemic. Everyone who had the opportunity to work remotely had this form ordered. In Kuwait, only 33% of employees worked remotely during the pandemic. More than 26% did not work remotely.

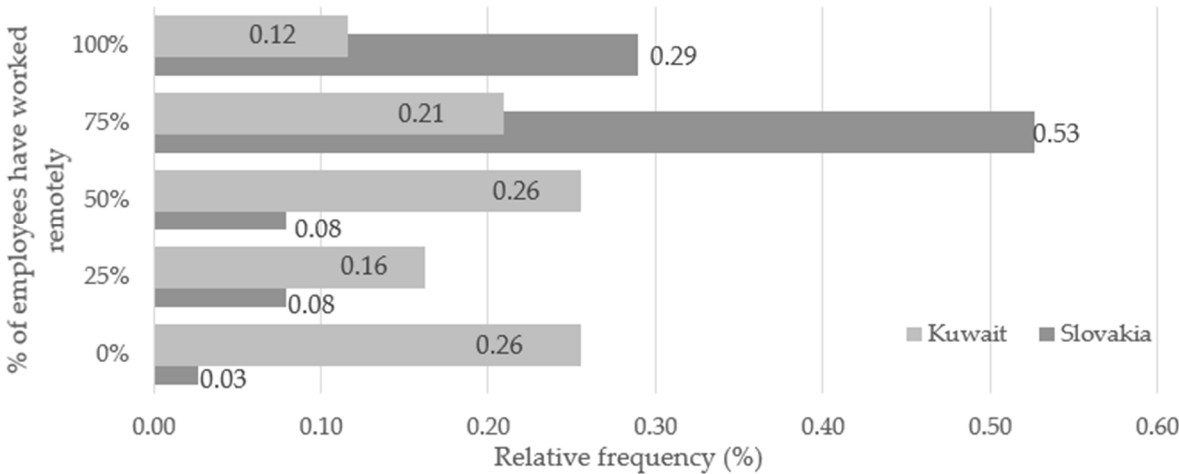

**Figure 3.** Relative remote work frequency.

Figure 4 shows the number of days that employees were allowed to work remotely during the pandemic. In Slovakia, the most available options were to work 3 days or 5 days a week outside the office. In this case, the dependence was that the higher the number of employees working from home, the more days could employees use to work online outside the office. In Kuwait, the most available options were to work 3 days or 5 days a week outside the office. In this case, almost 26% of employers did not work remotely. There was no significant dependence because the variance of options was too large and employers had a more balanced mix of offices.

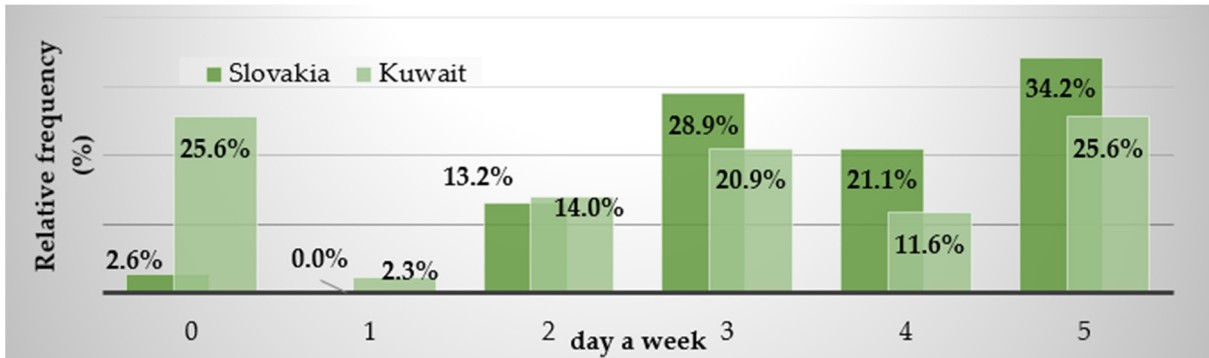

**Figure 4.** Remote work possibility.

Employers were asked to make statements about remote work. Figure 5 shows the statements of employers about remote work. Employers in Slovakia increased the level of remote work (26.3%), but 31.6% preferred limited remote schedules with fixed rules. Employers in Kuwait (30.2%) declared it necessary to return to the office, but 14% declared that the office space was completely unnecessary in combination with remote work. Responses from our survey showed that almost 58% of employers in Slovakia did not want to proceed with redundancies and reductions in the number of employees, but rather see this form as an opportunity to reduce operating costs at an increased or permanent rate of remote work. Employers in Kuwait have admitted redundancies (14% of responses), seeing an opportunity in investing in technology and developing online or IT services (48.8%).

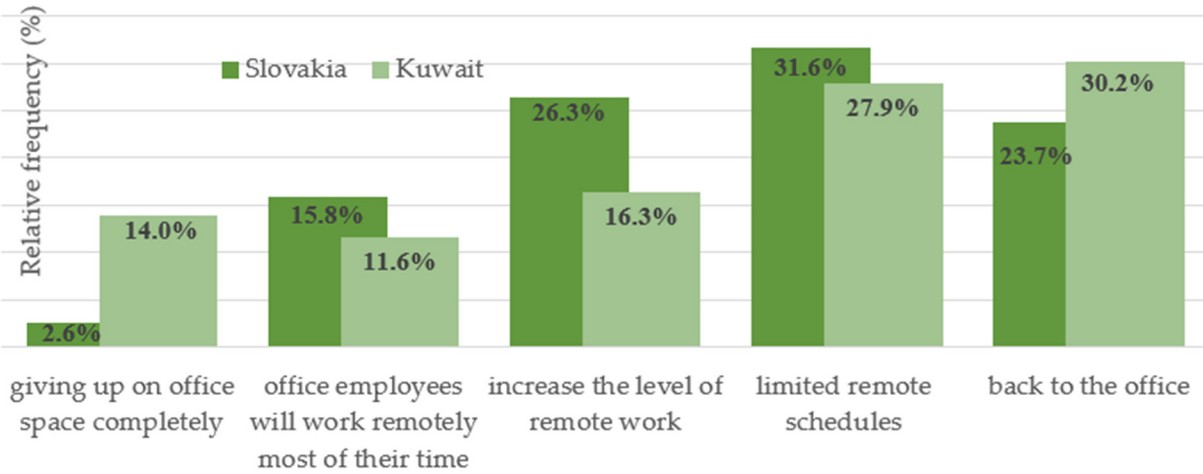

**Figure 5.** Remote work statements.

Employers were asked to identify three barriers to remote work in their organization. Many organizations learned to work in the online space and use technology during the lockdown. The most significant barriers to remote work in Slovakia mentioned by employers were:

(1) Face to-face contact is required (71%)
(2) Internet and other IT related issues (47.3%)
(3) Mental health and well-being of employees (39.4%)

In the government and public services sectors, it was common to work in a closed office using technology or to be in direct contact during meetings. The move from the office to the home was widespread. Therefore, the second significant barrier was that many households did not have sufficient internet coverage, which made work difficult. The long lockdown and low socialization acted as barriers that decreased productivity and corporate manners. Employers wanted to disrupt the routine of working from home as much as possible with new incentives to protect the health and well-being of employees.

The same question was asked in Kuwait. The most significant barriers to remote work in Kuwait mentioned by employers were:

(1) Face to-face contact is required (55.8%)
(2) Fairness, as not all employees could benefit from remote working due to nature of their work (44.2%)
(3) Physical presence to operate equipment is required for monitoring performance of employees (both 37.2%)

Fairness was the second most frequently cited barrier that employers did not want to or could not deal with. Since the nature of the work did not allow everyone to work outside the workplace, the employers followed the rules that, in such a case, there will be no one.

Employers considered access to the workplace necessary due to the availability of documents and equipment, without which the quality and productivity of the work performed decreases. The equality and non-disadvantage approach have helped to maintain the work ethics and well-being of employees during the period of government action. Thus, it was clear that based on the responses of employers from various sectors, such as oil and mining, a physical presence was required for the operator even during the lockdown.

*4.3. Work Environment and Office Space Settings after the Pandemic*

Employers were asked to rank team scenarios after the pandemic. In Slovakia, as shown in Figure 6a, full teams working remotely full-time (52.6%) or part of the time (44.7%) and part of the team working remotely full-time (50%) was not acceptable. One of the most acceptable answers (21%) was the possibility of a hybrid work model, where part of the team worked remotely part of the time (3+ days/week). In principle,

working from home will become a regular part of the work option because it is mostly acceptable for employers. Occasionally, teams will have to meet face-to-face as an essential part of work. After a thorough analysis, employers will allow employees to work part of the time remotely, once or twice a week, and a presence of 2 or 3 days at the workplace will be required. The most preferred option was full-time work in the office (26.3%).

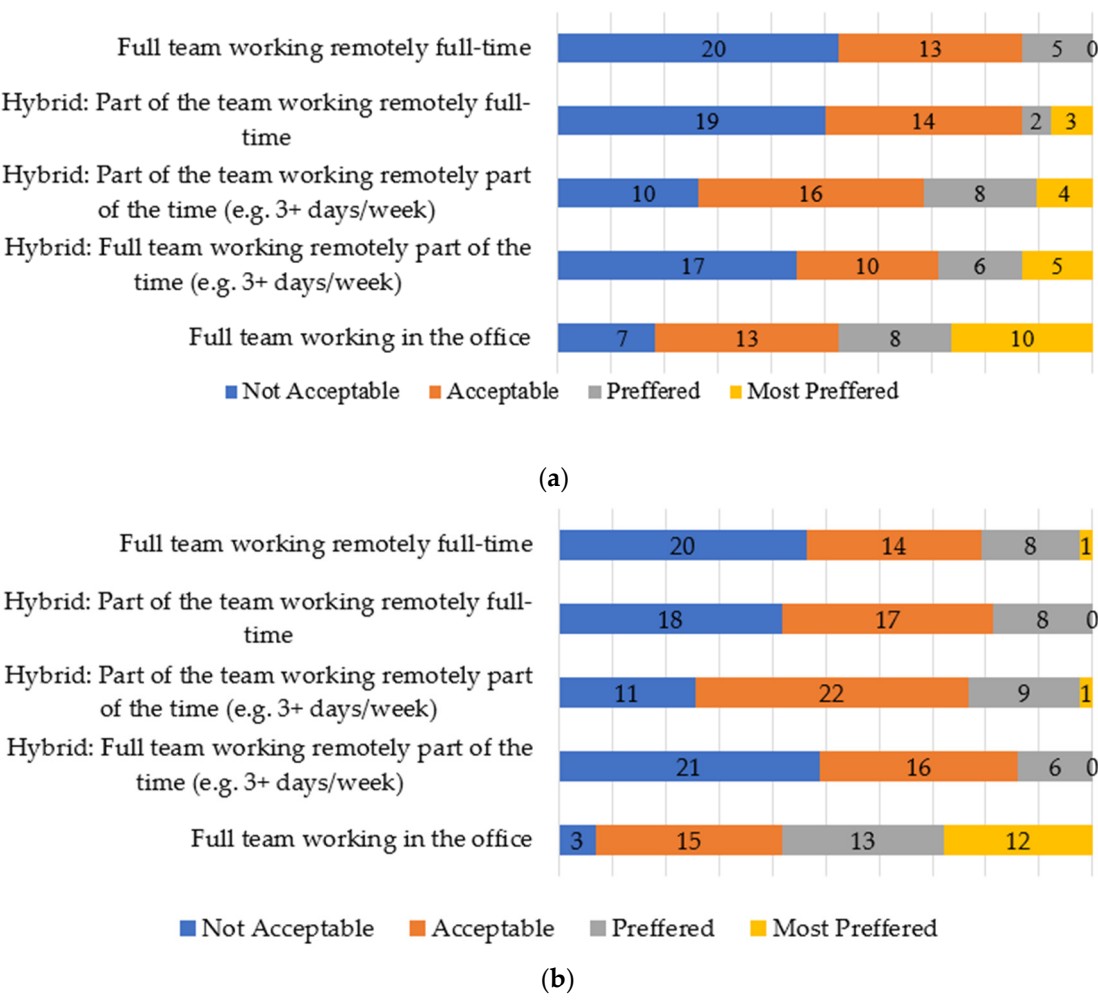

**(a)**

**(b)**

**Figure 6.** (**a**) Team preferences in Slovakia, (**b**) team preferences in Kuwait.

In Kuwait, as shown in Figure 6b responses were similar to Slovak employers' preferences. The most acceptable answer (20.9%) was the possibility of a hybrid work model, where part of the team works remotely part of the time (3+ days/week). More employers preferred full-time work from the office (18.6%) or a hybrid work model when part of the team works remotely part of the time (18.6%). However, the results showed that the transition to a purely remote way of working is unacceptable (46.5%).

These two countries had similar conclusions for the private sector. When looking for connections, it was mainly the case that private organizations wanted to make their premises safer because coworking or complete open space is useless in this situation. Is the end of the open office coming? The most popular office layout would still be an open office. However, the open office would have high partitions, which can be considered one of the elements of a new flexible office. Employers primarily attempt to use all the space, save money and protect employees. This office layout and the activity-based office layout support the greater flexibility of employees and remote work. Regular team rotation or work in smaller groups could support a hybrid work model to maintain employee well-being, ensure sustainable productivity and promote organizational culture and personal contact.

## 5. Discussion

### 5.1. Theoretical Contribution

The work environment will change in the near future (RQ1). It is important to note that the reconstruction of the current office space involves a demanding and very expensive workplace layout for employers. The reconstruction may be financially unprofitable or completely impossible [49]. According to respondents in Slovakia (65.7%) and Kuwait (48.8%), a major reconstruction or change of office space is not expected to occur in the next 3 years it can be assumed that employers realize that if the pandemic worsens, employers can move work activities online [50].

However, in order to make the "old" office more attractive, respondents in Slovakia (almost 53%) will upgrade hardware and equipment. On the other side, respondents in Kuwait (34.9%) will import some of the home comforts (planting, soft furnishing) and support healthier lifestyles with a relaxing area, juice bar or small gym space.

Future offices should have the 3 E's—an economic office with higher value for employers and employees [51], an environmental office with focus on green and sustainable technology [52,53] and an efficient office with a suitable, accessible and work effective layout [54]. The office layout should optimize safety, comfort, and functionality. The construction of new office spaces (Green Office) based on sustainability, ecological standards and research studies [3,55] has important impacts on the attractiveness and occupancy of the modern type of office space [1,56,57].

### 5.2. Practical Implication

The attitude of employers toward the home office was perceived at the level of crisis resolution during the pandemic constraints (RQ2). Wohlers et al. [58], in their research, described the home office as the main goal to increase employee productivity, creativity and cooperation. It should be noted that some sectors did not have problems switching to a full online regime. Whereas employers in the ICT sector have not been so severely affected by the crisis [59], employees working for employers in other sectors have been forced to reconsider their approach to managing employees and adapt their perception of office and work environment [60,61].

In the first wave, the pandemic measures shut down all offices for several weeks except essential positions. If this was not the case, the remote work or a hybrid work model was used worldwide [41]. Consequently, certain changes occurred in terms of the use of premises. In the case of permanent remote work, respondents in Slovakia (65.7%) and respondents in Kuwait (58.1%) stated that they would not have unused office space despite unclear pandemic developments in the short term. However, the office layout will significantly change in the short or medium term. The question is whether some changes will not be only temporary because of the regulations.

In the term run, employers do not plan to change their office layouts significantly. However, this statement contradicts with other studies [17,62,63] that have indicated the need for adjustments. It can be argued that the possibility of remote work results in the lack of, or insufficient reorganization of, office space. Yet, remote work has opened new possibilities for situations that previously seemed unsolvable. One of the possibilities could be use of office type defined as a new flexible office. This type of office has all the elements of the abovementioned approaches (see Figure 7) and can respond quickly to the needs of employers and employees. In addition, it incorporates other elements of open office workspace, coworking workspace and activity-based workspace, which makes the work environment more dynamic. It accommodates the equipment of individual offices and flexibility of premises leading to a flexible office, making it ready for any quick adjustments and changes that might be required when an unpredictable situation occurs. The great benefit is that it combines online and on-site space [12], and thus supports a hybrid work model.

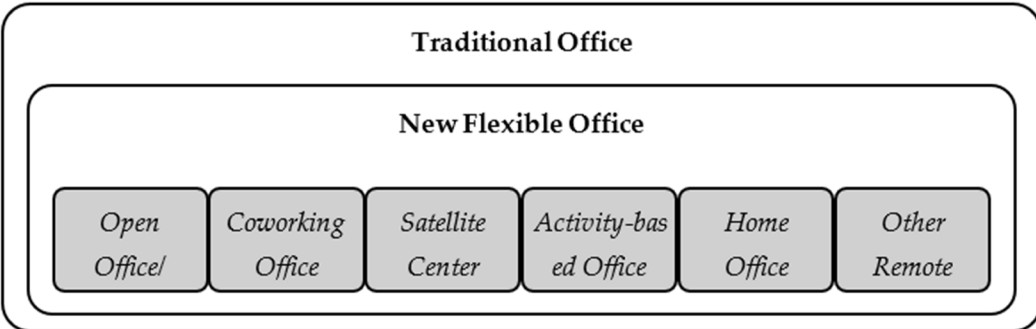

**Figure 7.** Transformations of the office setup in the hybrid work model. Source: Authors' contribution.

*5.3. Managerial Implication*

Based on this research, up to 85% of respondents' answers in Slovakia and almost 68% of respondents' answers in Kuwait returned employees back to their offices once the pandemic measures were lifted. When employees returned from remote work to office-type work, a number of questions remained open [64]. What needs to change?

The key to unlocking the individual and organizational high-performance opportunities of the future is the design [65], ability to control the environment [51] and development and implementation of a new safe workplace model [18]. Subsequently, office pandemic measures were taken to reduce office movement [38]. Typical features were the rotation of teams, the transfer of many essential activities to the online space (such as communication) and the attendance system [66,67].

In order to solve the issue of access of employees to the office, employers can apply the three approaches mentioned below in Table 3—the 6- Feet Office, More Remote Work or Sustainable Space [64]. However, according to the results, a very common response was that nothing would change after the pandemic. Many employers, primarily in the government and public services sectors, will not respond to changes in office space settings in the future. In our results, in the short-term, almost 63% of respondents' answers in Kuwait and almost 87% of respondents' answers in Slovakia did not confirm changes in real estate strategy. In the private sector, plans were intended to consolidate the office space into one business location using elements of the open office workspace, coworking workspace and activity-based workspace, which would help work environment become more dynamic and meet the diverse needs of employees.

**Table 3.** Office spaces after employees return.

| 6-Feet Office | More Remote Work | Sustainable Space |
|---|---|---|
| **The Same Size Only Fewer People at Once** | **Employers Will Support Work from Home** | **Employers Will Start Reducing Their Office Space** |
| The current size of the offices will remain unchanged. | Without offices, everything takes place online and outside the common areas. | The workplace will no longer be a single place, but an ecosystem of different places to support the safety, functionality and quality of working conditions of employees. |
| Flexible working conditions, fewer people in the office at once, sufficient distance at the social distance in the office. | Transition from office work to telework, formally defined in contracts. | |
| | Employers will pay for the equipment and work needs of employees. | |
| Continued work in the office HO as a benefit | From Classic Worker to Mostly Teleworker | Hybrid Work Model/Mix |
| Fixed Desk | Share Desk Policy/Flexible Desk | |

Source: Authors Contribution.

After the end of the protective measures and the pandemic, everything will return to "normal," but organizations with a low level of occupational safety and health management will not improve much [68]. However, due to the unclear end of the pandemic, employers must adhere to the set of measures. The future workplace must be digital, less hierarchical and more flexible [69]. The solution is the 6-feet office as a model and a new standard typical for office space [50]. The 6-feet office model is characterized by typical features, such as greater distance between desks, alternating teams in open space offices, regular disinfection and a clean climate are a matter of course.

### 5.4. Hidden Findings

We investigated employer's perspectives of where and in what settings the work will be performed in the post-pandemic time. Specifically, we discussed the changes employers will apply in terms of the work environment and office layout.

Some sectors did not change their philosophy as they depend on a physical office because of the clients and the services they offer. In the sectors of the government and public services, traditional office spaces—enclosed, shared (by 2–5 employees)—were utilized for a long time before pandemic. The results are clear in this area, and employers do not plan to change anything in the future in Slovakia or Kuwait. However, the layout of the office has undergone changes, at least in the case of the open office, which is the least suitable for the current situation and protection of employees. Several architectural studios are working on a new, ideal office layout. It will not take long, and instead of using handles, doors will open automatically. Upon arrival to work, employees' temperature will be measured automatically. All these elements will define the *Flexible Office Space*—a space that can be changed or adjusted with a sufficient distance between individual worktables, separate zones and high-quality, clean and divided air conditioning.

Informal variables that might have influenced the results are cultural differences and lifestyle [70]. In Slovakia, frequent and occasional commuting to the office or workplace is very common. Hence, employers will more likely support a hybrid work model where the workplace location will not be necessarily defined. On the other hand, commuting in Kuwait seems necessary. Since majority of employees in Kuwait live together with an extended family, working from home may have negative effect on their performance.

### 6. Conclusions

What is the future of offices? It can be argued that, in principle, offices will not disappear. The findings suggest that an increasing mobile workforce and expansion of the new work style will not mean an office exodus but will certainly have an impact on office utilization. From this point of view, we can summarize our findings as successful in terms of changing the approaches of employers towards office space because of COVID-19.

What to do next? In essence, the pandemic has become the driving force behind changes and the speed of applying flexible elements to work. A short-term approach led to restrictions and adjustments of premises due to the established measures. However, each organization will have to evaluate what flexible elements and to what extent they apply. According to employers, medium-term and long-term approaches are unsettled or still undecided. Only the time and the development of the situation will be the driving force for further changes. The highest priority will be protecting employees and their health, connecting employee health with satisfaction and efficiency and creating a balance between private and professional life. For the contribution of the authors, three possible solutions have been proposed for managers, which can be used in the approach to office space.

There are also positive outcomes associated with the COVID-19 pandemic. Elements of flexibility and flexible office space, which were inadmissible in the past, are now at least part of benefits. Employers offer these benefits as part of increasing employee protection and comfort. Online meetings have been set up, online education has started and localization restrictions have disappeared. The term locality has lost its meaning because new technologies have opened new possibilities at the global level. These possibilities will

clearly not disappear in the future. In particular, employers that have used the benefits of flexible offices before plan to keep them today and in the near future, or expand their layouts with telecommunications and online rooms.

Some employers have found that working from home is more suitable, while others are still looking for a compromise and reach for coworking spaces. However, if employees have the chance to choose their option, it will depend on the goals of the organization. It can be assumed that the number of job positions working remotely even after the pandemic will continue to depend on the nature of the job.

The shift from defined office space to a new model and functionality is possible. Premises can be divided into smaller spaces or modified according to the needs of individual departments in organizations. In the case of Slovakia, it will be the introduction of remote work from an occasional benefit to a regular option in the form of a new flexible office. In the case of Kuwait, it will be the development of new environmental office space. Equally important factors in individual countries are culture, customs, traditions and lifestyles, which might explain changes in attitudes toward the use of office space in the future [71,72]. When hybrid work model seems to be the key to success for some organizations, remote work starts to become a traditional work setting. It is important to emphasize that by losing office space, we can lose the main aspects that add additional value to work: socialization and the feeling that work makes sense.

### 6.1. Future Research Directions

COVID-19 has affected all industries worldwide. The world began to recover from the initial shock during the spring of 2020. It turned out to be a suitable solution to go into an online environment that was able to replace the work environment and offices. We encourage future researchers in this topic to compare specific sectors in different regions or countries and find common features, dependencies or differences in the attitudes of individual employers [73–76].

Based on small sample of respondents ($n = 38$ for Slovakia, $n = 43$ for Kuwait), we cannot generalize the research results to the broader population of Kuwait and Slovakia. For a very deep and specific output, it would be necessary to approach every single organization and examine its corporate culture and rules for office politics in Slovakia and Kuwait. Therefore, future research is needed to support the findings and conclusions.

### 6.2. Research Limitations

An important limitation was the time shift. The survey was anonymous and accompanied by a cover letter. The return was reduced because data collection occurred partly during the summer, when respondents were on vacation or there was general fatigue on the subject of the pandemic. The online survey was distributed in two countries during 2021, when many employers overcame the shock of the new situation and had new procedures in place. Time lag and other types of restrictions could have led to a more subjective perception of measures and changes in approaches to office space.

The second limitation was an uneven, small sample of respondent's answers ($n = 38$ for Slovakia, $n = 43$ for Kuwait). In this case, we only addressed managers in organizations that had the prerequisite or declared real experience in the use of remote work and flexibility in the workplace before COVID-19. Many employers did not want to disclose their experience and knowledge, and the answers were not sufficiently useful in this research.

**Author Contributions:** Conceptualization, M.B. and D.A.S. All authors have read and agreed to the published version of the manuscript.

**Funding:** This research received no external funding.

**Institutional Review Board Statement:** This study was conducted in compliance with the regulations of the national level Act no. 18/2018 on personal data protection and amending and supplementing certain Acts, and in compliance with the regulations of the internal regulation No. 16/2017 Rector's Directive Comenius University in Bratislava Full reading of the internal regulation No. 23/2016

Rector's Directive Comenius University in Bratislava, which issues the Code of Ethics of the Comenius University in Bratislava. All participants were acquainted with the use of the answers and by filling in the answers, they agreed to their further use. All responses were anonymous.

**Informed Consent Statement:** Not applicable.

**Data Availability Statement:** The datasets used during the current study are available from the corresponding author on reasonable request.

**Conflicts of Interest:** The authors declare no conflict of interest.

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
