# Peer review of "Offices after the COVID-19 Pandemic and Changes in Perception of Flexible Office Space"

_sustainability, doi:10.3390/su141811158_

Round 1

Reviewer 1 Report (Previous Reviewer 3)

Dear authors

There is a clear effort in writing this research, so thank you, but there are some notes please consider:

1. Defining the problem of the study accurately and clarifying what is the gap in this study?

2. What is the researchers' justification for the lack of hypotheses in this study?

3. The sample size is small in both countries. Why?

4. Determine the type of sample used.

5. The statistical analysis is simple with no hypotheses.

Best Regards

Author Response

Dear reviewer,

Thank you for report. We send our responds on suggestion.

Reviewer 2 Report (Previous Reviewer 4)

The authors have successfully implemented all my comments. I consider the paper worthy of publication.

Author Response

Dear reviewer,

Thank you for report. 

Reviewer 3 Report (Previous Reviewer 2)

The article is interesting, the researched problem has scientific potential, and the literature review represents a strength. However, some problems need to be solved:

1. Reference numbering does not follow the instructions for authors.

2. The use of self-administered questionnaires can generate a problem that may affect the relevance of the research: the bias effect or common method bias - CMB (see: Podsakoff PM, MacKenzie SB, Lee, JY, Podsakoff NP. Common method biases in behavioral research: A critical review of the literature and recommended remedies. Journal of Applied Psychology. 2003; 88(5):879-903.). Such problems arise when data on independent and dependent variables emanate from the same respondent, and the same measurement scale exists throughout the questionnaire. Authors must take action to prevent common method bias - CMB (e.g., https://doi.org/10.3390/ijerph182312387)

3. Data processing is performed using descriptive statistics. Considering that you have a data set based on a questionnaire, the paper would gain value if complex statistical methods were used to establish the relationships between variables (SEM, MANOVA, multiple regressions, etc.)

The article presents scientific value and can be published after carefully reviewing the reported issues.

Author Response

Dear reviewer,

Thank you for report. We send our responds on suggestion.

Round 2

Reviewer 3 Report (Previous Reviewer 2)

The paper can be published in current form.

This manuscript is a resubmission of an earlier submission. The following is a list of the peer review reports and author responses from that submission.

Round 1

Reviewer 1 Report

Comments to authors

Overall, I find that this is a qualitative study not quantitative; hence, the writing should be explained correctly. Although author used quantitative research testing (logical induction) and some descriptive analysis, the sampling was not drawn properly from population and cannot be used to make generalization. Hence, authors need to revise the paper in major correction. In anyway, similar research was already done; in fact, authors did not mention the dissimilarities with these researches:

i)               de Lucas Ancillo, A., del Val Núñez, M. T., & Gavrila, S. G. (2021). Workplace change within the COVID-19 context: a grounded theory approach. Economic Research-Ekonomska Istraživanja, 34(1), 2297-2316.

ii)             Yang, E., Kim, Y., & Hong, S. (2021). Does working from home work? Experience of working from home and the value of hybrid workplace post-COVID-19. Journal of Corporate Real Estate.

iii)           Babapour Chafi, M., Hultberg, A., & Bozic Yams, N. (2021). Post-pandemic office work: Perceived challenges and opportunities for a sustainable work environment. Sustainability, 14(1), 294.

iv)           Mayerhoffer, M. (2021). The impact of Covid-19 on coworking spaces: evidence from Germany. Journal of Corporate Real Estate, 23(3), 170-185.

1. Introduction

Overall, the introduction was written properly with few research problems. However, the writing can be improved. In the last paragraph (line 56), authors suddenly said that the research focus on two countries; in which, as reader, I don’t even know what is the objective of the research. Hence, I believe the research’s main objective/goal should be put in the last paragraph in Introduction section, not in Methodology section. I also understand some points of research problem; hence, the main research problem should be highlighted significantly/consistently with the main research objective.

2. Materials and Method

Overall, the writing can be understood well. However, I am not sure if authors cited appropriate theories/models from literature review. For example, if Figure 1 (line 62) was taken from previous scholars, hence, it can be seen that authors have cited appropriate literature review. However, if Figure 1 is from authors, hence, authors should explain/include theories/models related to the field of flexible offices. Please revise.

i)               Authors explained that Section 2 is about “overview of literature” (line 57-58), but the title is “2. Materials and Method”. I believe it’s actually Literature Review (line 62). Please revise.

ii)             Figure 1 (line 80) has no source; authors should put who suggest the framework and explain in the text. Is this framework invented by previous scholars or by authors?

iii)           In the last second paragraph (line 154), authors referred “both” but previous sentences mentioned only “public sector”. I believe authors should write one of the words as “private sector”. Please revise.

iv)           Sentences in last paragraph was not consistent with everything explained previously. From reading, I understand that the research problem is about “flexible office space” (just like title). However, in the last paragraph, suddenly the research problem is about green building? Really confusing and inconsistent, please revise.

3. Research Methodology

Overall, the writing can be understood; however, there are so many information were not provided; this is making the research methodology to be unconvincing. In fact, the writing did not follow how to report quantitative research correctly. These should be revised thoroughly.

i)               Authors claimed that the research was using quantitative (in abstract and line 181); however, so many words of “qualitative” was mentioned in the paper (line 175, 177, 463). I believe this is qualitative not quantitative research.

ii)             In quantitative study, it is very important to report who are the sample and population; and how sample was drawn from population. However, authors only mentioned employers from Slovakia and Kuwait. Where are they come from? In an organization? Anybody from these 2 countries? Is this using random or non-random sampling method? How many employers were involved in this research? This information is very important before authors can make conclusion that their respondents represent the Slovakia and Kuwait. In fact, in last paragraph you cannot simply claim that Kuwait represent the Arab model and Slovakia represent the European model (line 198-199) because this is not true; support your debate with literature review if this is true. There are so many countries in European and Arab that are totally different. Usually, in quantitative study, sampling frame is identified. - Suddenly, in the first paragraph in Section 4 (line 203), you let readers know your sample drawn from Slovakia were only 38 and from Kuwait were only 43 (Table 1); in which, the proportion and population for each country and organization were not explained. In fact, from Slovakia only one private sector involved and in Kuwait not sure if they were from private or public sector. Obviously, the sampling drawn was not appropriate, did not represent population, and was not on the basis random sampling; hence, results cannot be used to generalize the data.

iii)           There is no research objective that answering the research question. There were 3 research questions (line 169-172); however, no research objectives were constructed to solve the research questions. This looks like qualitative not quantitative research. In quantitative research, researchers must provide research objective to answer research question.

iv)           The writing of hypothesis must be revised. It can be several hypotheses to achieve only one research objectives; in which, the number of research hypotheses will usually be more than research objectives. However, there is no research objective in this paper; hence, it is really confusing what is actually the hypotheses is going to prove/achieve? Hypotheses will demonstrate how research objectives are achieved through significant test; however, the writing of hypothesis sentence was just not right. In fact, if the hypotheses were to answer research questions, why are they not consistent? Please revise. Additionally, in line 480, H2 was not tested using any test. Hypothesis is to test something using a test; H2 has no test, hence, it is not a hypothesis. It is not quantitative study at all.

v)             What is the instrument used in this research? Please explain because this is very important to understand either the result will be valid. It is better to provide list of questions in this section, not to surprise readers and reveal it in Results section. Please also explain the reliability and validity or psychometric property of instruments used in your research. If you are using structured questions for interview (that you took from qualitative study), you should also explain it because usually in qualitative study, researcher should also prove the reliability and validity of structured questions.

vi)           What is the research procedure? Please explain because this is very important to understand either the sampling is valid.

vii)         What is actually your research variables? It is very confusing because the hypotheses were not written correctly. Is it employers’ approach and attitude? Or is it the different countries, Slovakia and Kuwait? In quantitative study, the variables selected to be researched is determined.

viii)        Line 194, please revise your sentence. What do you mean by “H0 is rejected…”? Is it to reject hypothesis null? Please use full sentence not symbol. In fact, you didn’t explain what is your hypothesis null, then what you want to reject? In addition, all hypothesis should be tested whether to reject of to accept; why only H0 is tested?

ix)            In line 186, authors claimed to use “logical induction” to test hypothesis. In fact, in line 191, authors provide formula to do the testing.  Please cite a scholar that claim this is the right test to use, and the scholar’s name that provide the formula. Are these scholars agreed that your sampling method (that looks like purposive sampling) can be used for “logical induction” test? You are using very small number of respondents to represent Slovakia and Kuwait (see Table 1); are these scholars also agreed to the amount of sample? Have you test whether your data fall under parametric or non-parametric test? Do these scholars also agree with the statistical assumption in test selection suitable with your data? Please provide evident/proof.

4. Data Analysis and Results

Overall, the data analysis looks like results from qualitative study not quantitative study that readers can make generalization from the findings.

i)               The sampling was not drawn appropriately from population. I believe, authors used data from qualitative study to write the paper from quantitative perspective. Hence, it can be seen authors used purposive sampling; in which, results cannot be used to make generalization to the population. Authors might want to revise the whole paper that findings can only be used for the sample and not to conclude/make generalization for employers in Slovakia and Kuwait.    

ii)             Line 222, please always use “respondents in Slovakia/Kuwait” instead of “in Kuwait/Slovakia” because your results obviously did not represent employees from Kuwait/Slovakia rather than selection of few organizations. Please also revise the whole paper with same terminology.

iii)           Figure 2a, what is the 3 categories for bar chart? Please check thoroughly to avoid simple writing mistakes.

iv)           Line 331, why the word AFTER was in capital?

v)             Line 480, authors claimed that H2 is hypothesis, but no testing was done. Then, do not use the word “hypothesis”, use the word “objective”. This is obviously qualitative study not quantitative.

5. Discussion

Overall, authors only discuss their research results without relating with previous research findings that already been discussed in section 1 and section 2. Hence, it quit difficult to understand the contribution of this research as compared to previous research. Please revise.

6. Data Analysis and Results

Overall, conclusion represent the whole writing.

Author Response

Dear reviewer,

Thank you for report. We send our responds on suggestion:

  1. i)               de Lucas Ancillo, A., del Val Núñez, M. T., & Gavrila, S. G. (2021). Workplace change within the COVID-19 context: a grounded theory approach. Economic Research-Ekonomska Istraživanja, 34(1), 2297-2316.
  2. ii)             Yang, E., Kim, Y., & Hong, S. (2021). Does working from home work? Experience of working from home and the value of hybrid workplace post-COVID-19. Journal of Corporate Real Estate.

iii)           Babapour Chafi, M., Hultberg, A., & Bozic Yams, N. (2021). Post-pandemic office work: Perceived challenges and opportunities for a sustainable work environment. Sustainability, 14(1), 294.

  1. iv)           Mayerhoffer, M. (2021). The impact of Covid-19 on coworking spaces: evidence from Germany. Journal of Corporate Real Estate, 23(3), 170-185.
  • We added mentioned research articles to discussion or to other part in our research

  1. Introduction

Overall, the introduction was written properly with few research problems. However, the writing can be improved. In the last paragraph (line 56), authors suddenly said that the research focus on two countries; in which, as reader, I don’t even know what is the objective of the research. Hence, I believe the research’s main objective/goal should be put in the last paragraph in Introduction sectionnot in Methodology section. I also understand some points of research problem; hence, the main research problem should be highlighted significantly/consistently with the main research objective.

  • The aim of the paper has been moved and explained at the end of the Introduction
  • The objective and of the paper to be well highlighted in the Introduction
  • Literature review has been revised and more recent and relevant sources used
  • We added and revised 30+ references to the "to the point", more from years 2018-2022, some older valuable were also added

  1. Materials and Method

Overall, the writing can be understood well. However, I am not sure if authors cited appropriate theories/models from literature review. For example, if Figure 1 (line 62) was taken from previous scholars, hence, it can be seen that authors have cited appropriate literature review. However, if Figure 1 is from authors, hence, authors should explain/include theories/models related to the field of flexible offices. Please revise.

  • This Figure 1 was moved to discussion / source authors based on discussion with authors was added
  1. i)               Authors explained that Section 2 is about “overview of literature” (line 57-58), but the title is “ Materials and Method”. I believe it’s actually Literature Review (line 62)Please revise.
  • Rewritten

  1. ii)             Figure 1 (line 80) has no source; authors should put who suggest the framework and explain in the text. Is this framework invented by previous scholars or by authors?
  • This Figure 1 was moved to discussion / source authors based on discussion with authors was added

iii)           In the last second paragraph (line 154), authors referred “both” but previous sentences mentioned only “public sector”. I believe authors should write one of the words as “private sector”. Please revise.

  • Hole section rewritten
  1. iv)           Sentences in last paragraph was not consistent with everything explained previously. From reading, I understand that the research problem is about “flexible office space” (just like title). However, in the last paragraph, suddenly the research problem is about green building? Really confusing and inconsistent, please revise.
  • Hole section rewritten, green building is results, not isssue

  1. Research Methodology

Overall, the writing can be understood; however, there are so many information were not provided; this is making the research methodology to be unconvincing. In fact, the writing did not follow how to report quantitative research correctly. These should be revised thoroughly.

  1. i)               Authors claimed that the research was using quantitative (in abstract and line 181); however, so many words of “qualitative” was mentioned in the paper (line 175, 177, 463). I believe this is qualitative not quantitative research.
  • This research is more quantitative – online survey data collection
  1. ii)             In quantitative study, it is very important to report who are the sample and population; and how sample was drawn from population. However, authors only mentioned employers from Slovakia and Kuwait. Where are they come from? In an organization? Anybody from these 2 countries? Is this using random or non-random sampling method? How many employers were involved in this research? This information is very important before authors can make conclusion that their respondents represent the Slovakia and Kuwait. In fact, in last paragraph you cannot simply claim that Kuwait represent the Arab model and Slovakia represent the European model (line 198-199) because this is not true; support your debate with literature review if this is true. There are so many countries in European and Arab that are totally different. Usually, in quantitative study, sampling frame is identified. - Suddenly, in the first paragraph in Section 4 (line 203), you let readers know your sample drawn from Slovakia were only 38 and from Kuwait were only 43 (Table 1); in which, the proportion and population for each country and organization were not explained. In fact, from Slovakia only one private sector involved and in Kuwait not sure if they were from private or public sector. Obviously, the sampling drawn was not appropriate, did not represent population, and was not on the basis random sampling; hence, results cannot be used to generalize the data.
  • Description added in chapter "methodology"

iii)           There is no research objective that answering the research question. There were 3 research questions (line 169-172); however, no research objectives were constructed to solve the research questions. This looks like qualitative not quantitative research. In quantitative research, researchers must provide research objective to answer research question.

  • Description added in chapter "methodology" and discussion
  •  
  1. iv)           The writing of hypothesis must be revised. It can be several hypotheses to achieve only one research objectives; in which, the number of research hypotheses will usually be more than research objectives. However, there is no research objective in this paper; hence, it is really confusing what is actually the hypotheses is going to prove/achieve? Hypotheses will demonstrate how research objectives are achieved through significant test; however, the writing of hypothesis sentence was just not right. In fact, if the hypotheses were to answer research questions, why are they not consistent? Please revise. Additionally, in line 480, H2 was not tested using any test. Hypothesis is to test something using a test; H2 has no test, hence, it is not a hypothesis. It is not quantitative study at all.
  • After your thorough review we decided to remove the Hypotheses as they have been irrelevant to the objectives of the study
  • No statistic measures and hypothesis after revision
  • Descriptive data analysis was completed using rankings and percentages for scale. 4-point Likert scale questions were recorded and converted to numerical. See Table 1
  • We chose the non-random sampling method for the needs of the research with a small sample for individual categories.
  1. v)             What is the instrument used in this research? Please explain because this is very important to understand either the result will be valid. It is better to provide list of questions in this section, not to surprise readers and reveal it in Results section. Please also explain the reliability and validity or psychometric property of instruments used in your research. If you are using structured questions for interview (that you took from qualitative study), you should also explain it because usually in qualitative study, researcher should also prove the reliability and validity of structured questions.
  2. vi)           What is the research procedure? Please explain because this is very important to understand either the sampling is valid.
  • Description added in chapter "methodology" based on literature and studied authors contribution

vii)         What is actually your research variables? It is very confusing because the hypotheses were not written correctly. Is it employers’ approach and attitude? Or is it the different countries, Slovakia and Kuwait? In quantitative study, the variables selected to be researched is determined.

viii)        Line 194, please revise your sentence. What do you mean by “H0 is rejected…”? Is it to reject hypothesis null? Please use full sentence not symbol. In fact, you didn’t explain what is your hypothesis null, then what you want to reject? In addition, all hypothesis should be tested whether to reject of to accept; why only H0 is tested?

  • After your thorough review we decided to remove the Hypotheses as they have been irrelevant to the objectives of the study
  1. ix)            In line 186, authors claimed to use “logical induction” to test hypothesis. In fact, in line 191, authors provide formula to do the testing.  Please cite a scholar that claim this is the right test to use, and the scholar’s name that provide the formula. Are these scholars agreed that your sampling method (that looks like purposive sampling) can be used for “logical induction” test? You are using very small number of respondents to represent Slovakia and Kuwait (see Table 1); are these scholars also agreed to the amount of sample? Have you test whether your data fall under parametric or non-parametric test? Do these scholars also agree with the statistical assumption in test selection suitable with your data? Please provide evident/proof.
  • Descriptive data analysis was completed using rankings and percentages for scale. 4-point Likert scale questions were recorded and converted to numerical. See Table 1
  • We chose the non-random sampling method for the needs of the research with a small sample for individual categories.

  1. Data Analysis and Results

Overall, the data analysis looks like results from qualitative study not quantitative study that readers can make generalization from the findings.

  1. i)               The sampling was not drawn appropriately from population. I believe, authors used data from qualitative study to write the paper from quantitative perspective. Hence, it can be seen authors used purposive sampling; in which, results cannot be used to make generalization to the population. Authors might want to revise the whole paper that findings can only be used for the sample and not to conclude/make generalization for employers in Slovakia and Kuwait.
  • Based on small sample of respondents (n=38 for Slovakia, n=43 for Kuwait) we cannot generalize research results to a broader population of Kuwait and Slovakia. Therefore, future research is needed to support the findings and conclusions / added to subchapter 6.1 and 6.2
  1. ii)             Line 222, please always use “respondents in Slovakia/Kuwait” instead of “in Kuwait/Slovakia” because your results obviously did not represent employees from Kuwait/Slovakia rather than selection of few organizations. Please also revise the whole paper with same terminology.

iii)           Figure 2a, what is the 3 categories for bar chart? Please check thoroughly to avoid simple writing mistakes.

  1. iv)           Line 331, why the word AFTER was in capital?
  • Yes, mistake / corrected
  1. v)             Line 480, authors claimed that H2 is hypothesis, but no testing was done. Then, do not use the word “hypothesis”, use the word “objective”. This is obviously qualitative study not quantitative.
  • After your thorough review we decided to remove the Hypotheses as they have been irrelevant to the objectives of the study

  1. Discussion

Overall, authors only discuss their research results without relating with previous research findings that already been discussed in section 1 and section 2. Hence, it quit difficult to understand the contribution of this research as compared to previous research. Please revise.

  • Added contribution and implications / discussed with authors worldwide

Reviewer 2 Report

The article is interesting, and the researched problem has scientific potential. However, some problems need to be solved: 

1. Literature review should include more recent sources (2018-2021) and be enriched with relevant references. 

2. The use of self-administered questionnaires can generate a problem that may affect the relevance of the research: the bias effect or common method bias - CMB (see: Podsakoff PM, MacKenzie SB, Lee, JY, Podsakoff NP. Common method biases in behavioral research: A critical review of the literature and recommended remedies. Journal of Applied Psychology. 2003; 88(5):879-903.). Such problems arise when data on independent and dependent variables emanate from the same respondent and the same measurement scale exists throughout the questionnaire. Authors must take action to prevent common method bias - CMB (e.g., https://doi.org/10.3390/ijerph182312387) 

3. Data processing is performed using descriptive statistics. The article would gain value if complex statistical methods were used to establish the relationships between variables (SEM, MANOVA, multiple regressions, etc.) 

4. There is a need for a discussion section built in the context of dialogue with researchers in the literature review. 

5. In my opinion, a section of conclusions that includes theoretical and managerial implications, along with research limitations and future research directions, would be helpful. 

The article presents some scientific value and can be published after carefully reviewing the reported issues. 

Author Response

Dear reviewer,
Thank you for report. We send our responds on suggestion:

  1. Literature review should include more recent sources (2018-2021) and be enriched with relevant references.
  • Literature review has been revised and more recent and relevant sources used
  • We added and revised 30+ references to the "to the point", more from years 2018-2022, some older valuable were also added
  1. The use of self-administered questionnaires can generate a problem that may affect the relevance of the research: the bias effect or common method bias - CMB (see: Podsakoff PM, MacKenzie SB, Lee, JY, Podsakoff NP. Common method biases in behavioral research: A critical review of the literature and recommended remedies. Journal of Applied Psychology. 2003; 88(5):879-903.). Such problems arise when data on independent and dependent variables emanate from the same respondent and the same measurement scale exists throughout the questionnaire. Authors must take action to prevent common method bias - CMB (e.g., https://doi.org/10.3390/ijerph182312387)
  • coordinated with article Podsakoff PM, MacKenzie SB, Lee, JY, Podsakoff NP and added to references
  • A two-Stage SEM by authors Bocean et. al. was not used in research
  1. Data processing is performed using descriptive statistics. The article would gain value if complex statistical methods were used to establish the relationships between variables (SEM, MANOVA, multiple regressions, etc.)
  • Based on small final sample size, these methods were not used / no relevant results were drawn / not applicable.
  1. There is a need for a discussion section built in the context of dialogue with researchers in the literature review.
  • On this point, discussion was rewritten and relevant articles added. 
  • Discussion was rewritten as discussion with relevant authors from introduction and literature review plus with relevant older / newer articles
  1. In my opinion, a section of conclusions that includes theoretical and managerial implications, along with research limitations and future research directions, would be helpful.
  • Chapter 5 Research Findings and Discussion contains subchapters implications and contribution in theoretical, practical and managerial background plus subchapter hidden findings which are based on experiences and skills of authors and cultural surface.

Reviewer 3 Report

Dear authors,

Thank you for your great effort in this paper

There are some notes:

- The introduction needs more explanation and should be woven like a story and not as separate paragraphs.

-Hypotheses are not written in this way like a master's or doctoral thesis, but must be developed with theoretical review.

- The population and sample of the study must be accurately determined, and the type of sample used as well as the sample size must be determined.

- Where is the source of the questionnaire?

- What is the justification for relying on the two countries of Slovakia and Kuwait?

- The statistical analysis used is very simple.

- The results were not discussed, nor was the theoretical contribution and practical implications of the research written.

-Limitations and future research should be added.

- References are few, more should be added.

Author Response

Reviewer 3

Dear reviewer,

Thank you for report. We send our responds on suggestion:

The introduction needs more explanation and should be woven like a story and not as separate paragraphs.

  1. Introduction has been revised and elaborated.
  2. Literature review was rewritten - characteristics of countries added.

-Hypotheses are not written in this way like a master's or doctoral thesis, but must be developed with theoretical review.

 After your thorough review we decided to remove the Hypotheses as they have been irrelevant to the objectives of the study.

- The population and sample of the study must be accurately determined, and the type of sample used as well as the sample size must be determined.

  1. Description added in chapter "methodology"

- Where is the source of the questionnaire?

  1. Self-designed questionnaire based on the studied literature
  1. Added in chapter "methodology"
  1. Not all data from questionnaire were used and connected with research.

For Slovakia:

https://docs.google.com/forms/d/e/1FAIpQLSeXr-3MgdHhBaxCKGWH7zQ_z3BcZ7IAJ1GYfUY44uBfZv379g/viewform

For Kuwait:

https://docs.google.com/forms/d/e/1FAIpQLScI6pzAwb0XbIhZhImV-ZGkPdbAaYEbOYaGhAfkSqiLT53vQw/viewform 

- What is the justification for relying on the two countries of Slovakia and Kuwait?

Both authors are from Slovakia - gained PhD. in flexibility in Slovakia and Kuwait. Author 1 is living in Slovakia and dealing with this topic, author 2 is living and working in Kuwait. Also economic similarities as population, area were key factors. Both countries dealing with similar problems during COVID-19 - no legislative. Kuwait has not developed research in this area, Slovakia has gained some legislative forms in last period. 

We tried to find similarities and diferrencies in managers approach based on hidden factors and cultural variables.

- The statistical analysis used is very simple.

  1. We used Cramer, Pearson chi-square test, ANOVA and even more, but finding were not very interesting in this way, because of size of sample was incorrectly subdivided. 

- The results were not discussed, nor was the theoretical contribution and practical implications of the research written.

  1. Yes, added and rewritten - Research Findings and Discussion. Contributions were added in discussion and conclusion, chapter 5 was split in four parts as implications.

-Limitations and future research should be added.

  1. Added in conclusion as part 6.1. Future Research Directions; 6.2. Research Limitations

- References are few, more should be added.

  1. We added and revised 30+ references to the "to the point", more from years 2018-2022, some older valuable were also added

Reviewer 4 Report

This paper presents a simple statistical analyses based on data from an online survey to investigate: i) employers’ willingness to change or adapt to demands in the post-pandemic era; ii) employers’ attitudes towards future plans and changes in office layout in the post-pandemic time. The study compared two countries – Slovakia and Kuwait – with two different approaches to the office solution.

 In general, I find the paper quite interesting, even though the analyses are based on very simple statistical methods, such as frequency tables, graphs, and contingency tables. More interesting is the way the author(s) organised the research questions.

 I have some suggestions.   

 1) The aim of the paper would be better explained at the end of the Introduction. I would like the objective and contribution of the paper to be well highlighted in the Introduction.

2) The authors tried to draw more general conclusions based on small samples (n=38 for Slovakia, n=43 for Kuwait), without discussing the limitations due to the low number of interviewees. I invite the authors to discuss this point.

3) In Section 2, the authors discussed traditional and flexible types of office configuration, without addressing the general labour market context of the countries. I invite the authors to introduce the relevant economic and labour market scenario (please, see Castellano et al., 2016; 2019 and Su et al., 2022 below).

4) The authors have introduced some theoretical aspects, but it is still unclear how the variables were selected. The author(s) should better describe the data sources. I would like to see the questionnaire and how the variables were selected.

5) I find the discussions limited to a few lines at the end of the paper (Section 6). Most of the results are discussed simply from a statistical point of view, whereas they can have some potential in terms of policy implications and measures. I would suggest that the author(s) broaden the discussion on how their results can be useful to policy makers or society at large. The discussion of practical policy proposals should allow the paper to gain in originality.

Additional references:

 Castellano, R., Manna, R., Punzo, G. (2016). Income inequality between overlapping and stratification: a longitudinal analysis of personal earnings in France and Italy. International Review of Applied Economics, 30(5), 567-590.

Castellano, R., Musella, G., Punzo, G. (2019). Exploring changes in the employment structure and wage inequality in Western Europe using the unconditional quantile regression. Empirica, 46(2), 249-304.

Su, Z.F., Fu, Y.Z., Chen, M.Y. (2022). Impacts of a Gender Ratio Change on China’s Wage Income Distributions. Emerging Markets Finance and Trade, 58(7), 2066-2078.

Author Response

Dear reviewer,

Thank you for report. We send our responds on suggestion:

1) The aim of the paper would be better explained at the end of the Introduction. I would like the objective and contribution of the paper to be well highlighted in the Introduction.

  1. The aim of the paper has been moved and explained at the end of the Introduction
  2. The objective and of the paper to be well highlighted in the Introduction

2) The authors tried to draw more general conclusions based on small samples (n=38 for Slovakia, n=43 for Kuwait), without discussing the limitations due to the low number of interviewees. I invite the authors to discuss this point.

  1. Added in conclusion as part 6.1. Future Research Directions; 6.2. Research Limitations
  2. Description added in chapter "methodology"
  3. Chapter 5 and 6 - Research Findings and Discussion. Added subchapters and rewritten -  Contributions were added in discussion and conclusion, chapter 5 was split in four parts as implications / contributions
  4.  

3) In Section 2, the authors discussed traditional and flexible types of office configuration, without addressing the general labour market context of the countries. I invite the authors to introduce the relevant economic and labour market scenario (please, see Castellano et al., 2016; 2019 and Su et al., 2022 below).

  1. This chapter was completely changed. We added some relevant and market factors, which could help reader to better understand this topic.
  2. Su et al., 2022 - added
  3. We added and revised 30+ references "to the point", more from years 2018-2022, some older valuable were also added

4) The authors have introduced some theoretical aspects, but it is still unclear how the variables were selected. The author(s) should better describe the data sources. I would like to see the questionnaire and how the variables were selected.

  1. Description added in chapter "methodology"
  2. After your thorough review we decided to remove the Hypotheses as they have been irrelevant to the objectives of the study
  3. No statistic measures and hypothesis after revision
  4. Descriptive data analysis was completed using rankings and percentages for scale. 4-point Likert scale questions were recorded and converted to numerical. See Table 1
  5. We chose the non-random sampling method for the needs of the research with a small sample for individual categories.
  1. For Slovakia - questionnaire:

https://docs.google.com/forms/d/e/1FAIpQLSeXr-3MgdHhBaxCKGWH7zQ_z3BcZ7IAJ1GYfUY44uBfZv379g/viewform 

  1. For Kuwait- questionnaire:

https://docs.google.com/forms/d/e/1FAIpQLScI6pzAwb0XbIhZhImV-ZGkPdbAaYEbOYaGhAfkSqiLT53vQw/viewform  

5) I find the discussions limited to a few lines at the end of the paper (Section 6). Most of the results are discussed simply from a statistical point of view, whereas they can have some potential in terms of policy implications and measures. I would suggest that the author(s) broaden the discussion on how their results can be useful to policy makers or society at large. The discussion of practical policy proposals should allow the paper to gain in originality.

  1. Added in conclusion as part 6.1. Future Research Directions; 6.2. Research Limitations
  2. Chapter 5 Research Findings and Discussion contains subchapters implications and contribution in theoretical, practical and managerial background  plus subchapter hidden findings which are based on experiences and skills of authors and cultural surface.
  3. Discussion was rewritten as discussion with relevant authors from introduction and literature review plus with relevant older / newer articles
  4. There is a limitation, based on small sample of respondents from both countries. 

Based on small sample of respondents (n=38 for Slovakia, n=43 for Kuwait) we cannot claim that Kuwait represent the Arab model and Slovakia represent the European model and more representative researches need to be added in future. -Added

Round 2

Reviewer 1 Report

Comments to authors:

1)     Introduction

a.     Authors keep arguing that their research is new and is solving research gap. But I cannot understand, what is the need to compare “flexible offices” in Slovakia and Kuwait at the first place? For example, is Slovakia a first world country and Kuwait is third world country until there is a need to compare between them? Researchers usually organize research when there is research issue or problem that need to be solved. However, I do not see any research problem. Authors need to justify about this to make clear justification whether the research findings will be significant or not.

b.     Line 69. Quantitative study should have objective to be achieved not questions to be answered. It is not wrong to have questions in quantitative study, but it is wrong for not having objective to be achieved in quantitative study.

2)     Literature Review section was very short and insufficient.

a.      Research should be based on strong theoretical field of study especially for quantitative study; however, authors did not explain related theories and previous research works with the main research focus (flexible offices). Authors should explain related theories in the field of study that usually referred by researchers/scholars to understand the basic phenomenon of “flexible offices”. I thought “Figure 1” in previous article draft was taken from established theory/model of “flexible offices”, however it was not but was conceptualized by authors. Then, on what basis authors did their research?

b.     Literature review related to “flexible offices” should be discussed and synthesized before organizing the research. However, authors did not really discuss about it. If this is not done, how authors can be sure that they have been done research that: (i) is different from previous? (ii) can add significant contribution from what have been done? (iii) to conclude that other countries have similarity and dissimilarity with what they found in Kuwait and Slovakia? (iv) can be useful to be integrated with findings from previous research? There is so much related research were done using different countries. Authors should mention about it so that it can be compared to their findings.

c.      Section 2.1 was written in one paragraph. Please revise.  

3)      Research Methodology should be revised

a.     Authors must have strong understanding in quantitative research methodology. Line 206, authors claim that “50 private and public organization” are the sampling frame. In don’t think authors really understand what is referred to “sampling frame”. Authors want to make quantitative research using sample from Slovakia and Kuwait. Firstly, you need to know how many public and private organizations in these countries; this is called as population. It is later called as “sampling frame” when you chose you sample using this population. But you did not know how many of them instead of purposively selected any organizations to become your sample. “50 private and public organization” is not “sampling frame” but is your “sample”. In fact, your unit analysis is not organization but individual human that come from these 50 organizations. Then, you must know the total number of them within each organization; you do not simply choose them if you really have sampling frame. Another concern is, which state in either Slovakia or Kuwait are these respondents came from? Does your sample represent all the states? Please read and understand more about research methodology for quantitative study. Misunderstanding about it is making the article writing become odd; it just not right.

b.     Line 208, what do you meant by “refer to Chapter 2”? Are you expecting readers to read what you have read or expecting readers to make literature review like you did? Article written by you should be stood alone. This is confusing, because you are making your paper to be dependent to another research or article. You do not expect readers to read another article just to understand you article; in which, it just not right.  

c.      Sampling method and sample size is different aspect. Obviously, your sampling method is purposive. However, your sample size also was insufficient to make conclusion and generalization to represent Slovakia and Kuwait. Line 201 is like “barking up the wrong bush” when you do not understand what is the importance of sampling in quantitative study. Again, this is just not right; do not cite other scholars when they are saying something else, but you are using it for something else. It is not wrong to have purposive sampling and insufficient sample size in quantitative study; just admit about it and follow the limitation of the results implication.

d.     Line 210, are you sure that “...non-probabilistic sampling…” is “…used in assuring that small groups of samples are adequately represented”? This is serious research methodology error!

e.     I had raised the issue of questionnaire validity and reliability in previous comments. However, no correction was made.

4)     Results

a.     For Figure 1a., I already commented that there are 3 different graphs for Slovakia and Kuwait, but no description what those graphs categories are represented. I do not see any correction was made.

5)     Research Findings and Discussion

a.     “Results” and “Research Findings” are the same. I think Section 5 should only be Discussion.

b.     Line 396 and 399 was not consistent. Authors said that respondents claimed that “…will not take place” in line 396. Inconsistently, line 399 said that “…will be expanded by 15%”. Please revise.

c.      I cannot accept that authors used the term as “Slovakia” and “Kuwait” from section “Discussion” until “Conclusion” because obviously sampling was done using purposive sampling; in which, findings did not represent the whole “Slovakia” and “Kuwait”. Authors should change it into “respondents from Slovakia/Kuwait”. It is wrong to generalize in discussion and conclusion that authors’ research findings represent the population of Slovakia and Kuwait.

d.     Line 405, I do not agree with the statement that future office should only have “…the key in 3E -…”. It is not proven yet only these 3 are the keys; it might have more than this. Therefore, authors should refer to theories/models related to “flexible offices” that already discussed by previous scholars; in which, authors did not discuss about this in section 2 (Literature Review). A strong basis of theoretical must be referred to justify about this. For example, see this research:

Yu, R., Burke, M., & Raad, N. (2019). Exploring impact of future flexible working model evolution on urban environment, economy and planning. Journal of Urban Management, 8(3), 447-457.

https://reader.elsevier.com/reader/sd/pii/S2226585618302140?token=F697DA224106CD88241A0251D0046B4BEDD2D320C68D8CB587E2F3756C000D34AD34A9E4C1EC149C1EF5711FD4D2180B&originRegion=eu-west-1&originCreation=20220708151010

e.     I do not see what have been discussed in section 5.1 is something that contribute to theoretical contribution. Please refer to my comment in 2(b). How do you know that your research finding is new when you did not discuss what have been researched by previous researchers? Don’t you think similar research was done using more rigorous research method? See this research:

(i)             Research using London samples:

Forbes, S., Birkett, H., Evans, L., Chung, H., & Whiteman, J. (2020). Managing employees during the COVID-19 pandemic: Flexible working and the future of work. https://kar.kent.ac.uk/85918/1/managerial-experiences-during-covid19-2020-accessible.pdf

(ii)            Research using German samples:

Gauger, F. (2021). New Work Environments: The Economic Relevance of Flexible Office Space. https://tuprints.ulb.tu-darmstadt.de/19705/1/Dissertation_Jan_Felix_Gauger_final.pdf

(iii)          Research using Finland samples:

Gauger, F. (2021). New Work Environments: The Economic Relevance of Flexible Office Space. https://tuprints.ulb.tu-darmstadt.de/19705/1/Dissertation_Jan_Felix_Gauger_final.pdf

f.      I cannot understand why suddenly Figure 7 appeared. Either the model is from previous researchers or authors, it should be put in section 2 (literature review).  It is clearly stated by author in line 187 that this research was quantitative research. Then, how can you put your qualitative findings (Figure 7) in section Discussion? I think authors should understand thoroughly how quantitative research is reported.

g.     Line 459, last sentence in first paragraph in section 5.3 is written without academic support; but the sentence is a fact that should be supported by citation. In fact, previous scholars have agreed that not all jobs can be done from home. Do not simply make conclusion that it is applied to all employees around the world across all job positions.

h.     I do not understand why Table 4 is suddenly appeared in section Discussion? It is not related at all with results. How come it was discussed as findings? How come suddenly the “6 feet office” was discussed; in which, it has no relation at all with results and research question?

i.      Line 497. It is odd to have hidden findings in quantitative study. Why don’t you changed it to mix method of qualitative and quantitative study? It is more relevant.

6)     Conclusion

a.     Line 526. I think it is close-minded to conclude that “…offices will not disappear so soon.” It’s like giving clue that it will disappear sooner or later. Inconsistently, authors cited by themselves that only 37% of job can be done at home (line 56).

b.     Line 536-537. It is not right to claim that authors examined respondents’ changes in perception of attitudes before, during and after the pandemic. In quantitative study, longitudinal research should be done by collecting data using different time frames to claim a conclusion like this.

c.      I do not think that conclusion was written to represent the research findings, but it was written using something else. For example, in line 554, suddenly some managers are looking for “coworking” and “compensation packages”? this is not the results. Why is suddenly the conclusion?

d.     Line 580, authors agreed that they cannot generalized findings to population of Slovakia and Kuwait. However, in section “Results” and “Findings and Discussion” you keep arguing that your findings applied for those in Kuwait and Slovakia. You should change it to “respondents”.

7)     Is this paper is considered new when authors already put in this website?

https://www.preprints.org/manuscript/202206.0294/v1

8)     References

Authors did not really used references they cited to explain, discussed, and synthesised the research topic thoroughly. For example, references number [40] was a research with similar topic done in German. However, authors did not compare what they found in their research findings and what have been found by others in other countries. In fact, many relevant, related, and updated research articles were not cited.

Reviewer 2 Report

The paper can be published in current form.

Reviewer 3 Report

Approved

Reviewer 4 Report

The authors understood most of my comments well. A few perplexities remain.

The first concerns the choice of sampling design which, being non-probabilistic, does not allow the results to be generalised, in probabilistic terms, to the entire population. The authors mention this problem only marginally (page 5, lines 208-209), but in my opinion, it should be better emphasised in the conclusions as well, as it constitutes a serious limitation of the work, undermining its methodological rigour.

Second, I would suggest that the authors also take into consideration the other two bibliographic references suggested in the previous review round when asking about the general labour market context of the countries.

Third, I consider it appropriate to include the questionnaire administered in the Appendix.